# The Influence of Local Antibiotic Therapy on the Microbiological, Clinical, and Radiological Outcomes Following Minimally Invasive Periodontal Surgery in the Treatment of Intrabony Defects—A Randomized Clinical Trial

**DOI:** 10.3390/antibiotics14090850

**Published:** 2025-08-22

**Authors:** Anna Skurska, Amelia Baczewska, Robert Milewski, Piotr Majewski, Radosław Charkiewicz

**Affiliations:** 1Department of Integrated Dentistry, Medical University of Bialystok, ul. M. Skłodowskiej-Curie 24A, 15-276 Bialystok, Poland; 2Department of Periodontal and Oral Mucosa Diseases, Medical University of Bialystok, ul. Waszyngtona 13, 15-269 Bialystok, Poland; amelia.baczewska@umb.edu.pl; 3Department of Biostatistics and Medical Informatics, Medical University of Bialystok, ul. Szpitalna 37, 15-295 Bialystok, Poland; robert.milewski@umb.edu.pl; 4Department of Microbiological Diagnostics and Infectious Immunology, Medical University of Białystok, ul. Waszyngtona 15A, 15-269 Białystok, Poland; piotr.majewski@umb.edu.pl; 5Center of Experimental Medicine, Medical University of Bialystok, ul. M. Skłodowskiej-Curie 24a, 15-276 Białystok, Poland; radoslaw.charkiewicz@umb.edu.pl

**Keywords:** periodontal regeneration, MIST, periodontal intrabony defects, local drug delivery

## Abstract

**Objectives**: Comparison of clinical, radiological, and microbiological outcomes following periodontal regeneration procedures with or without local antibiotic therapy. **Methods**: Forty patients, each presenting with a single vertical defect, were randomly assigned to either the test (SRP+ANB+MIST/M-MIST) or the control group (SRP+MIST/M-MIST). The periodontal regenerative procedures were carried out according to the general minimally invasive surgical technique principles, and the vertical bone defect was filled with an enamel matrix derivative (EMD—Emdogain^®^). Periodontal condition assessments were performed two weeks before the procedure, on the day of the surgical procedure, and at follow-up visit after 6 months. Radiographs were taken two weeks before, and 6 months after the surgery. To determine the microbiological profile of the surgical sites, subgingival plaque samples were collected for PCR analysis. **Results**: In both groups, a statistically significant pocket depth (PD) reduction and clinical attachment level (CAL) gain were observed over the 6-month follow-up period. The difference between the groups for PD and CAL parameters at 6 months was not statistically significant. Both groups showed a statistically significant reduction in the radiological depth and width of intrabony defects. Microbiological analysis revealed a statistically significant difference between the groups two weeks after subgingival antibiotic application for *Fusobacterium nucleatum*, *Tannerella forsythia*, and *Prevotella intermedia*. **Conclusions**: Periodontal tissue regeneration procedures according to minimally invasive principles (MIST/M-MIST) with the use of EMD lead to improvements in clinical and radiological parameters. Local antibiotic application results in a reduction in bacterial counts in short-term observations. Its use prior to regeneration procedures does not lead to additional improvements in clinical and radiological parameters.

## 1. Introduction

Periodontitis is a chronic, multifactorial disease characterized by progressive destruction of the supporting tissues of the teeth [1]. Characteristic clinical symptoms include increased probing depth (PD), clinical attachment loss (CAL), and radiologically visible alveolar bone loss (RBL). Alveolar bone loss is classified as horizontal and vertical defects, including intrabony defects and craters [2,3]. The consequence of the disease is a number of functional and aesthetic disorders that worsen the patient’s quality of life [1,4].

The etiopathogenesis of the disease involves complex interactions between microorganisms and the host immune response. It is induced by dysbiotic dental plaque biofilm. Disturbances of the ecological balance result in the selection of pathogenic microorganisms, leading to periodontitis [5]. Initially, Socransky et al. identified the so-called red and orange complex bacteria, associated with the progression of inflammation [6]. The development of sequencing techniques has enabled a more precise classification of periopathogens such as: *Porphyromonas gingivalis—P.g.*, *Tanarella forsythia—T.f.*, *Trepoema denticola—T.d.*, *Fusobacterium nucleatum—F.n.*, *Prevotella* spp., and *Aggreggatibacter actinomycetemcomitans—A.a.* [7,8,9,10,11,12].

The infectious nature of the disease directs therapeutic methods. The basis of periodontitis treatment is mechanotherapy—removal of supragingival and subgingival deposits using hand and ultrasonic instruments. Additional therapies include antimicrobial agents, immunomodulators, and systemic and local antibiotics [13,14,15,16].

Although numerous studies have confirmed the effectiveness of systemic antibiotic therapy in the treatment of periodontitis, an important aspect of its use is the development of bacterial resistance [17]. The side effects of antibiotic use in the treatment of not only periodontal diseases are also important. The results indicate that patients are at risk of developing allergies (5%), nephritis (3%), hematological problems (2–2.5%), gastrointestinal problems (5.5%), nervous system disorders (2%), and skin allergy symptoms (5.5%). This encourages the limitation of their use and the development of alternative treatment methods [18].

Local application of antimicrobial drugs in the treatment of periodontal diseases allows for achieving high therapeutic concentrations without systemic side effects and the risk of developing resistant strains. Local drug delivery systems (LDDSs) include antibiotics and antiseptics applied in various forms such as fibers, gels, strips, microspheres, liposomes, or nanoparticles [19]. The main indications for their use include single pockets ≥6 mm resistant to mechanical treatment, non-surgical treatment of periimplantitis, and preparation of the surgical site before periodontal tissue regeneration [20,21]. Recently published clinical guidelines for non-surgical treatment of periodontitis, in terms of local antibiotic therapy, showed a statistically significant greater PD reduction and CAL gain after the use of LDDSs [22].

In the case of advanced disease, surgical treatment should be considered [22]. Modern periodontal surgery focuses on the following regenerative techniques aimed at rebuilding periodontal structures: root cement, alveolar bone, and functionally anchored periodontal ligament fibers [23,24].

Regenerative surgery is indicated in case of intrabony defects with the coexistence of pockets ≥ 6 mm after non-surgical treatment, provided that appropriate oral hygiene is achieved [25,26]. The key to the success of the procedure is healing through primary closure, clot stability, and ensuring space for regeneration [27,28]. For this purpose, guided tissue regeneration (GTR), enamel matrix derived proteins (EMD), and growth factors are used, often combined with bone substitute biomaterials [29].

The addition of enamel matrix derivatives (EMDs) to surgical techniques has allowed for further improvement of regenerative outcomes. EMDs support cell differentiation, expression of cytokines and osteogenic molecules, and have antibacterial properties [20,30,31]. Studies have shown that EMDs improve the effectiveness of periodontal regeneration, while reducing the frequency of complications compared to GTR, where wound dehiscence is quite common [32].

Difficulties in achieving primary wound closure initiated the development of papilla preservation techniques (PPTs), which were later transformed into the modified papilla preservation technique (MPPT) and simplified papilla preservation flap (SPPF) [33,34,35,36]. The traditional surgical approach of preparing large flaps evolved into the minimally invasive technique (MIST) and modified minimally invasive surgical technique (M-MIST) aimed at reducing surgical trauma and surgical access with limited tissue mobilization. Improvements in regenerative techniques focused on enhancing clot stability and providing space for healing [27,37,38].

Although the effect of LDDS use in non-surgical treatment of periodontitis and in traditional surgical techniques is well documented, there is a limited evidence for its application in the surgical area when planning minimally invasive treatment [22]. For this reason, it was hypothesized that the subgingival application of a local antibiotic before the MIST procedure could provide better control of inflammation by reduction in bacterial account and better healing after surgery.

Therefore the aim of the study was to compare microbiological, clinical, and radiological results after periodontal regeneration procedures using minimally invasive techniques, with or without the use of local antibiotic therapy in preoperative preparation.

## 2. Results

All 40 patients completed the 6 months of follow-up visits with no dropouts. Any adverse events were reported. Patients were equally distributed between the study and control group without differences according to age, surgical access, and intra-surgical defect depth and width. Table 1 shows characteristics of patients, with teeth and defects included.

### 2.1. Periodontal Clinical Parameters

Enrolled patients were characterized by good oral hygiene expressed by low full-mouth plaque score (FMPS < 20%). Over time, FMPS increased in both the test and control groups, with no differences between the groups. There was a significant difference in FMBOP in favor of test group at the second examination. In relation to the surgical site, there was a significant change in the value of PI in the test group and BOP in the control group in the 6 months of observation. No statistically significant differences were found between the groups.

In both groups, there was a statistically significant reduction in PD and gain in CAL during the 6-month follow-up period. The difference between the groups after six months of follow-up was not statistically significant.

The dimension of gingival recession (GR) increased in the test group in a statistically significant way. A significant difference in the parameter value between the groups occurred in the sixth month of observation.

The dimension of keratinized tissue remained stable throughout the observation period with no differences between groups. All data are presented in Table 2A,B.

### 2.2. Microbiology Findings

Analysis of microbiological evaluation showed significant changes in the values of genome copies per mL (gc/mL) of *Treponema denticola*, *Fusobacterium nucleatum*, *Porphyromonas gingivalis*, and *Prevotella intermedia* during the observation period in the test group. Two weeks after intra-pocket antibiotic administration, the difference between the groups was statistically significant in favor of the test group in relation to *Fusobacterium nucleatum* and *Prevotella intermedia* (*p* = 0.019 and *p* = 0.03, respectively).

Both the test and control groups showed a statistically significant reduction in the value of gc/mL (genome copies/mL) of *Tannerella forsythia* during the 3-month observation period. Two weeks after intra-pocket antibiotic administration, a statistically significant greater reduction was observed in the test group compared to the control group (1.17 × 10^4^ gc/mL and 2.33 × 10^5^ gc/mL, *p* = 0.026, respectively).

At the 3-month examination, statistically significant differences were observed in the value of gc/mL of *Aggregatibacter actinomycetemcomitans* bacteria between the test and the control group (3.96 × 10^6^ gc/mL and 1.38 × 10^5^ gc/mL, *p* = 0.002, respectively). Detailed real-time polymerase chain reaction (PCR) test results are presented in Table 3.

### 2.3. Radiological Findings

At the 6-month observation there was a significant reduction in radiological intrabony defect depth (R IDD) and width (R IDW) in both groups. There was no significant difference between the groups. Data are presented in Table 4.

## 3. Discussion

The results obtained showed a statistically significant reduction in PD and a gain in CAL in both groups in the 6-month follow-up period. The difference between the groups after six months was not statistically significant. Both in the test group and the control group at the 6-month follow-up a statistically significant reduction in the depth and width of intrabony defects assessed on the basis of RVG was demonstrated.

Microbiological analysis showed a statistically significant difference between groups two weeks after intra-pocket antibiotic administration. This difference concerned the bacteria: *Fusobacterium nucleatum*, *Tannerella forsythia*, and *Prevotella intermedia.* After three months of observation, a statistically significant difference between groups was shown only for the bacteria *Aggregatibacter actinomycetemcomitans*.

There are few studies available presenting microbiological changes after the use of local antibiotic therapy in regenerative periodontal treatment. Reddy et al. [39] assessed the microbiological profile in the sites after regenerative surgical procedures. The study noted a significant reduction in the number of *Aa* and *Pg* in both sites from the baseline to the 3-month follow-up. In the intergroup comparison, a significant reduction in the number of *Aa* and *Pg* was found in the test sites compared to the control sites in the 3-month postoperative period. Zucchelli et al. [40] compared the efficacy of local and systemic antimicrobial therapy in guided tissue regeneration (GTR). According to the study, local antibiotic administration is more effective than systemic administration in preventing membrane contamination, but does not lead to additional improvements in clinical parameters. The results obtained by Zucchelli et al. [40] are consistent with the results of our own research.

Jung et al. [41] evaluated the efficacy of minocycline combined with open flap debridement (OFD) in the treatment of patients with severe periodontitis and the effect of repeated minocycline administration in the postoperative period. Twenty patients received topical minocycline combined with OFD. The control group underwent OFD alone. Clinical improvement was demonstrated in both groups 6 months after surgery, but the reduction in PD and BOP was significantly greater in patients who received topical antibiotics. The plaque index was similar in both groups at each follow-up time point. The additional use of topical minocycline during flap surgery as well as in the maintenance phase (3 months after surgery) for the treatment of severe periodontitis was shown to have a beneficial effect in terms of reduced PD and BOP and gained CAL. It should be emphasized that the studies referred to the traditional surgical technique (OFD) and not minimally invasive techniques, and the fact that the antibiotic was used several times in the postoperative period. The differences between the cited studies and our results can also be explained by documented discrepancies in the results achieved with different surgical approaches [42].

Yusri et al. [43] in a systematic review and meta-analysis (nine publications) presented the effects of local antibiotic therapy during surgical periodontal treatment. The review included studies that used different surgical techniques (OFD, OFD with platelet-rich fibrin or bone substitute, modified Widman flap, and GTR with bone substitutes) and antibiotics (metronidazole, minocycline, doxycycline, tetracycline, and moxifloxacin). Similarly to our work, a significant CAL gain and PD reduction were observed in short-term observations up to 6 months. In contrast, statistically significant improvement in the BOP parameter was noted in the groups using a local antibiotic. This discrepancy may result from methodological differences. One aspect is the difference in the scope of the surgical procedure; the other is the method of parameter recording. In our study, the BOP index was assessed for the entire tooth, not for a single measurement point. Seven of the nine studies were designed as split-mouth interventions. On the one hand, this is a beneficial design where one patient also acts as a control, but on the other hand, when antibiotics are administered intra-pocket this may pose a risk of a “carry-over effect,” in which drugs administered on one side could affect the control site via oral fluids. Taking into account the gingival crevicular fluid production flow, we are not able to eliminate the phenomenon of releasing medicinal substances into the patient’s oral cavity. None of the studies included analyzed this effect, so the actual effect of antibiotic delivery could be different than that reported in the studies.

Aimetti et al. [44] in a recently published study evaluated the effect of locally administered doxycycline on the clinical and molecular parameters of inflammation in the area with an intrabony defect before a planned regenerative procedure. The study included 44 patients. Similarly to our study, the SRP procedure was performed in each patient two weeks before the surgical procedure. Then, the patients were randomly divided into two groups—test and control. In the test group, after the SRP procedure the antibiotic doxycycline (Ligosan^®^) was applied intra-pocket to the site of the planned surgical intervention. On the day of the procedure, the MIST procedure was performed in both groups using locally derived enamel matrix proteins (Emdogain^®^) and xenogeneic material (Bio-Oss, Geistlich Pharma AG, Wolhusen, Switzerland). In contrast to our study, all patients received systemic antibacterial treatment (amoxicillin with clavulanic acid 1 g for 6 days). Two weeks before the procedure and on the day of the procedure, the inflammatory index (BOP) and the cytokine level in the gingival fluid (GCF) were assessed at the surgical site. Then, 2 weeks after the procedure, the Early Wound Healing Index (EHI) was assessed. The results obtained showed that the number of sites with BOP+ on the day of the procedure was statistically significantly lower in the test group than in the control group. In our study, in 2-week observations a decrease in the BOP index value was also obtained in the test and control groups, but the obtained values were not statistically significant. No difference was found between the groups. The lack of statistically significant differences in the BOP index values may be attributed to the different methodology of testing this parameter in our own research compared to the cited work.

A statistically significant improvement in PD and CAL parameters obtained in both groups in our research confirmed the efficacy of microsurgical procedures. Minimal flap elevation, as well as the lack of vertical incisions, allow for the reduction in adverse events associated with the procedure. This is important both for the regeneration process and the patient’s intraoperative and postoperative comfort [27,45,46]. In the conducted study, healing ran without inflammatory complications in all cases. There was no wound dehiscence, flap or papilla necrosis, pathological exudate, edema, or hematoma in any of the patients. They also did not complain of significant pain in the postoperative period. Therefore, a comparable clinical effect was achieved in both groups, despite statistically significant differences in bacterial concentration between them.

The microbiological results did not translate into radiological results either. In both groups, a statistically significant improvement in both parameters (R IDD and R IDW) was demonstrated, with no difference between the groups. The results obtained in our study are consistent with the results of other authors. In the study conducted by Reddy et al. [39], 15 patients with at least two opposite vertical bone defects were included, where hydroxyapatite with or without moxifloxacin was used. A significant mean reduction in bone loss was demonstrated in both the test and control sites, with no significant differences between the sides in 6 months of observations [39]. Agarwal et al. [47] evaluated the results of regenerative procedures using bone substitute material with or without local application of doxycycline in the intrabony defect. Analysis of clinical and radiological parameters was performed at baseline, 3 months, and 6 months after the regeneration procedure. In both groups, a reduction in PD and an increase in CAL were observed over the 6-month period with no statistically significant differences between the groups. Both groups showed significant filling of the intraosseous defect after 6 months, but the difference between the groups was not significant. Despite methodological differences such as the surgical technique used or the use of bone substitute materials, the results of the cited studies agree with the results of our study in terms of improving clinical and radiological indicators, as well as the lack of differences between the groups in terms of the parameters tested.

The reasons for the lack of statistically significant differences between the groups can be found in many factors that determine the outcome of regenerative treatment. Despite the infectious nature of periodontal lesions, we cannot forget about other aspects that affect the healing process. Factors related to the patient (hygienic habits and general health), local conditions (type, morphology of the intrabony defect, endodontic status of the tooth, and tooth mobility), and factors related to the operator have an impact on the final outcome of the procedure [48]. In the presented study, the condition for inclusion in the procedure was to achieve optimal oral hygiene expressed by an FMPS below 20%. The FMPS refers to the patient’s determination to maintain proper oral hygiene. Although all patients underwent supragingival cleaning and oral hygiene instruction at each follow-up visit, the increased FMPS value indicates deficiencies in this area, which could have a secondary impact on the values of other clinical parameters. The patient’s general condition, lifestyle, and exposure to chronic stress are important factors that modify the healing process. The exclusion criteria in our study were general diseases that could affect the healing process, smoking, antibiotic therapy up to three months ago, as well as pregnancy and breastfeeding. However, we could not influence any changes in the patients’ general condition during the 6-month observation. The above should also be taken into account when analyzing the final results of the study.

Another important aspect of our study is the use of EMD as a means of supporting the regeneration process. Although procedures with EMD are sensitive, the risk of complications is lower than in the case of GTR procedures [49,50]. The use of enamel matrix derivative (EMD) is dictated by its impact on the regeneration process, safety, and rarely reported minor postoperative complications [51,52]. It is also worth emphasizing that EMD has antibacterial properties in vitro, as demonstrated in the literature [53,54,55]. Spahr et al. [54] suggest that EMD has a positive effect on the composition of bacterial species in the postoperative periodontal wound by selectively inhibiting the growth of periopathogens that could impede healing and reduce the effectiveness of regenerative procedures. The study showed a clear inhibitory effect of EMD on the growth of Gram-negative periodontal pathogens. A significant decrease in the number of *A. actinomycetemcomitans* (*p* = 0.012) was noted in the study after 24 h. The species *P. gingivalis* and *P. intermedia* also showed a clear decrease in growth in the presence of EMD. After 24 h, no live microorganisms could be detected. Interestingly, no significant inhibition of the growth of Gram-positive bacteria was observed. On the one hand, the described properties of EMD may be an issue in the context of the microbiological analyses performed in our study, but the fact of its utilization in both groups excludes the effect in favor of only one of them.

Taking everything into account, the advisability of using local antibiotic therapy before regenerative surgery should be considered. Despite its significant impact on the concentration of periopathogenic bacteria, it undoubtedly has its limitations. The problem arises when the infection is not localized but affects multiple sites, when proper debridement has not been performed before medication administration, when there is insufficient space for it, or when the medication has been administered incorrectly. A problem may also occur when periodontal lesions are not caused by infection, the infection is only a consequence of it, and it is caused by something else (e.g., root fracture, foreign body in the periodontal pocket) [56]. Financial considerations are also important. Although systemic antibiotic therapy is significantly cheaper, the previously mentioned side effects remain. Even though the clinical evidence for the benefits of systemic medications, for example, the combination of amoxicillin and metronidazole, is stronger than for local antibiotics, current treatment guidelines are restrictive regarding the use of systemic antibiotics [22,57]. The issues mentioned above, as well as the lack of differences in clinical and radiological results, call into question the validity of using local antibiotics before minimally invasive regenerative procedures. We definitely need studies with a more unified methodology conducted on a larger number of subjects to find the answer.

### Limitations

The undoubted limitation of our research is the size of the study group. It would be reasonable to conduct a similar study on a much larger number of patients, but due to financial and time constraints, this was not possible. These were also the main factors that we had to take into account when deciding on microbiological tests. An additional aspect is the lack of placebo in the study. Patients aware of being in the control group could feel less disciplined and therefore less motivated to follow the recommendations for maintaining ideal oral hygiene. Effective use of placebo would require intra-pocket application of a preparation indifferent to periodontal tissues. Another limitation was the inability to eliminate all factors influencing the periodontal regenerative potential. Smoking patients were not included in the study, but we do not know whether the qualified patients had smoked in the past. As confirmed by clinical studies, the periodontal response to periodontal treatment is the same as in non-smokers only after more than 10 years of quitting smoking [58]. Another limitation of our study is the lack of operator blinding, which can only be explained by personal and organizational limitations. A factor that could have influenced the bacteriological results was the use of Emdogain^®^ in both groups, which, as confirmed by clinical studies, also has antibacterial effects [53,54,59]. However, the use of the EMD in all patients excludes intentional action in only one of the studied groups. Moreover, we wanted to assess the validity of the use of local antibiotic therapy in a commonly used method of treating intrabony defects, which is the combination of minimally invasive techniques with enamel matrix derivative.

## 4. Materials and Methods

### 4.1. Sample Size

The primary study outcome was the change in the CAL. Sample size calculation was performed in order to detect a clinically significant difference of 1.0 mm in the CAL between the two therapeutic procedures. Type I error was set at 0.05 level and power at 0.80, assuming that the standard deviation (SD) was 1.0 mm. The required sample size was calculated to be 16 patients in each group (total 32 patients). In anticipation of patient dropout up to 20%, a total of 40 patients were planned for enrolment. Screening continued until a total of 40 patients (20 per group) were enrolled (Figure 1).

### 4.2. Study Population and Experimental Design

Forty patients diagnosed with periodontitis stage III (28 women and 12 men), aged 28–64, treated at the Department of Periodontal and Oral Mucosa Diseases, Medical University of Białystok, Poland, between October 2019 and November 2022, were included in the study. The study was designed as a single-center, randomized, prospective clinical trial.

The protocol of the study was registered in clinicaltrials.gov (NCT07027137), conducted in accordance with the Helsinki Declaration of 1975, as revised in 2013, and was reviewed and approved by the local ethical committee (Bioethics Committee of the Medical University of Białystok, Nr.: R-I-002/403/2019, date 26 September 2019). The study was designed as a single-center, randomized, prospective trial and was conducted in accordance with the CONSORT statement. Informed consent was obtained from all subjects involved in the study.

The inclusion criteria for patients were presence of an intrabony defect with a pocket depth (PD) ≥ 6 mm and a radiological defect depth of ≥3 mm and width of ≥2 mm; over 18 years of age; FMPS < 20% and FMBOP < 20%.

The exclusion criteria for patients were as follows: periodontal treatment within 3 months prior to the study; antibiotic therapy within 3 months prior to the study; smoking; presence of systemic diseases affecting periodontal healing (such as diabetes mellitus, immune deficiencies, and cancer diseases); pregnancy and breastfeeding.

Allocation of patients to test (SRP+ANB+SUR) and control (SRP+SUR) groups was performed according to a randomization table prepared by a statistician. Each participant had only one intrabony defect treated in the project.

### 4.3. Clinical Examinations

All included patients underwent the first step of treatment including oral hygiene instructions, professional mechanical plaque removal (PMPR), control of plaque retentive factors (i.e., removing filling overhangs), and discussing the risk factors of periodontal disease. Oral hygiene was reinforced at each following visit. All appointments were carried out by one operator (A.B.).

Clinical examination was performed using a periodontal probe (PCP UNC15, Hu Friedy, Chicago, IL, USA). Measurements were rounded to 0.5 mm. It was performed three times: before treatment, two weeks later (on surgery day), and 6 months after treatment by the same experienced and calibrated examiner. The examiner was blinded with respect to the treatment procedure performed (A.B.).

The following clinical parameters were measured for each tooth with intrabony defect: plaque index (PI) measured on 4 surfaces of the tooth, bleeding on probing (BOP) measured in 4 measuring points of the tooth, pocket depth (PD) from the gingival margin to the bottom of the sulcus, gingival recession (GR) from the CEJ to the most apical extension of gingival margin, and clinical attachment level (CAL) from the CEJ to the bottom of the sulcus. When the cemento-enamel junction was not detectable, the filling margins were taken as the reference point. The values of PD, CAL, and GR from the deepest measurement point were taken for analysis. Full-mouth plaque score (FMPS) [60] and full-mouth bleeding on probing (FMBOP) [61] were calculated as a percentage on the four surfaces of each tooth.

Also pre- and postoperative radiographs were taken using the long-cone parallel technique, with the customized bite positioner. There were two parameters analyzed on the radiographs: defect depth (RxD), the vertical distance between the bone crest and the site on the root surface at which the periodontium width was normal (in mm); and defect width (RxW), the horizontal distance between the root surface and bone defect margin in the most coronal part of the bone crest (in mm).

### 4.4. Investigator Calibration

The repeatability of the measurements taken by the investigator for periodontal parameters using the Bland–Altman method was evaluated. Five patients not included in the study were used for investigator calibration. The investigator evaluated 4 teeth, each with 6 measurement points in each patient, 48 h apart. Repeatability of the measurements by the clinician was good, and the error (bias) of the measurements was not statistically significant (confidence interval includes zero).

### 4.5. Clinical Procedure

The test group consisted of 20 participants (11 women and 9 men; aged 30–64 years) who underwent the scaling and root planing (SRP) procedure with the subsequent application of local antibiotics (10% doxycycline, Ligosan^®^, Kulzer GmbH, Hanau, Germany) into periodontal pockets at the site of the planned surgical intervention.

In the control group, also consisting of 20 people (17 women and 3 men; aged 34–52 years), the SRP procedure was performed alone.

The therapeutic procedure (SRP) was performed under local anesthesia (Septanest 100, Septodont, Paris, France), using an ultrasonic device (EMS Piezon, Tip PS, EMS, Nyon, Switzerland) and hand instruments (Gracey currettes (SMS), Hu-Friedy, Chicago, IL, USA) by one operator (A.S.). Local antibiotics were applied directly after SRP using factory-prepared cartridges.

After two weeks, regenerative treatment was performed. Following local anesthesia (Septanest 100, Septodont, Paris, France), intrasulcular incisions and preparation of mucoperiosteal flap was performed according to the principles of the papilla preservation technique [2,62] and minimally invasive techniques [27,28]. The debridement of bone defect was performed with the use of hand instruments (Gracey currettes, Hu-Friedy, Chicago, IL, USA) and ultrasonic devices (EMS Piezon Tip PS, EMS, Nyon, Switzerland). Once the debridement of the intrabony defect was completed, the depth and width of the defect were assessed using a periodontal probe. The depth of the defect (IDD—intrabony defect depth) was measured from the bottom of the bone defect to the top of the ridge, while the width (IDW—intrabony defect width) was measured from the lateral surface of the root to the inner wall of the bone defect. Subsequently, 24% EDTA (ethylenediaminetetraacetic acid, PrefGel^®^, Straumann Group, Basel, Switzerland) was applied to the root surface. EDTA was removed after two minutes by rinsing abundantly with saline, and enamel matrix derivatives (Emdogain^®^, Straumann, Switzerland) were applied to the root surface. The procedure was completed with mucoperiosteal flap reposition and stabilization by means of vertical modified mattress sutures (Ethilon 6.0, Johnson & Johnson Company, New Brunswick, NJ, USA). If tooth mobility was detected, it was splinted after surgery. Postoperative care consisted of 0.2% chlorhexidine rinses (Eludril, Pierre Fabre Laboratories, Paris, France) twice a day for 2 weeks. Patients were also instructed not to eat hard foods and to avoid vigorous tooth brushing at the surgical area for 2 weeks. Sutures were removed 14 days postsurgery.

Recall appointments were scheduled after 1, 2, and 4 weeks and then at 3 and 6 months. During recall appointments, supragingival plaque was carefully removed with a brush. Healing and possible complications (flap dehiscence; flap or papillae necrosis; suppuration; inflammation; as well as pain exacerbations) were monitored during the follow-up appointments. Photographs of surgical area were taken on every visit. The patient and the surgeon were not blinded in the protocol. Figure 2 presents a case treated in test group.

### 4.6. Microbiological Evaluation

In order to assess the microbiome of intrabony defect areas, subgingival plaque samples were collected for real-time polymerase chain reaction (PCR) assays before and after surgery. The material was obtained by isolating the site and then inserting a sterile paper point (size ISO 30) into the periodontal pocket for 30 s. Samples were placed in Eppendorf tubes with 200 μL of PBS, centrifuged (60 s), and frozen at −20 °C. The samples were analyzed for the presence of six periopathogenic bacteria: *Treponema denticola*, *Fusobacterium nucleatum*, *Tannerella forsythia*, *Porphyromonas gingivalis*, *Prevotella intermedia*, and *Aggregatibacter actinomycetemcomitans.*

The PeriodontScreen Real-TM PCR (Sacace Biotechnologies Srl, Como, Italy) test was used, which allowed for quantitative detection of bacterial DNA in real time using fluorescent probes. The test contains an endogenous control detecting human DNA, allowing for verification of the correctness of sample collection and preparation and the efficiency of the PCR. Analyses were performed on samples collected 2 weeks before, on the day of surgery, and 3 months after the procedure. Results are presented in gc/mL (genome copies/mL).

For the extraction of DNA from subgingival plaque samples, we employed a commercial CE IVD diagnostic DNA isolation kit, specifically the TANBead Nucleic Acid Extraction Kit (Bacteria DNA Auto Plate) on the Automated Nucleic Acid Extractor Maelstrom 4800 instrument (Taiwan Advanced Nanotech Inc., Taoyuan, Taiwan), following the manufacturer’s protocol. Subsequently, we conducted a qualitative and quantitative assessment of the isolated DNA. The measures were performed using spectrophotometric techniques on a NanoDrop 2000c instrument (Thermo Scientific, Waltham, MA, USA). Furthermore, the concentration of the DNA solutions was determined using a fluorometric method with a Qubit™ dsDNA HS Assay Kit (Thermo Scientific, USA). This assessment measures allowed us to determine the overall quality of the DNA samples, providing confidence in their suitability for downstream analysis.

### 4.7. Radiological Evaluation

The radiological examination (RVG) was performed using the right-angle technique using individual positioners made of plasticized wax (LuxaForm, DMG Chemisch-Pharmazeutische Fabrik GmbH, Hamburg, Germany), placed on a standard positioner and bitten by the patient immediately before exposure. The positioners were stored in controlled conditions (dry, dark, and at room temperature) to maintain dimensional stability throughout the study period. RVG images were taken two weeks before the procedure and six months later. The images were used to assess the width (RIDW) and depth (RIDD) of the intrabony defect. The analysis was performed using Planmeca Romexis 4.6.1. R software (SLS UMB license) by one qualified investigator (A.B.). Figure 3 presents performed measurements.

### 4.8. Statistical Analysis

In the statistical analysis normality of distribution was verified using the Shapiro–Wilk test. Normality of distribution of the analyzed quantitative variables was not found. When comparing quantitative variables without normality of distribution, the nonparametric Mann–Whitney U test was used in the case of comparing two independent groups and the Wilcoxon test in the case of comparing dependent groups (two time points). When comparing multiple dependent groups (multiple time points), the Friedman test was used. The Pearson Chi-square test of independence was used to check the relationship between nominal characteristics.

The results were considered statistically significant for *p* < 0.05. Statistica 13.3 (TIBCO Software Inc., Palo Alto, CA, USA) was employed for calculations.

## 5. Conclusions

Periodontal tissue regeneration procedures performed according to the principles of minimally invasive surgery (MIST/M-MIST) using EMD lead to improved clinical and radiological parameters. Intra-pocket antibiotics application causes a decrease in bacterial counts in short-term observations. Local antibiotics therapy before regeneration procedures does not lead to additional improvements in clinical and radiological parameters in short-term observations.

## Figures and Tables

**Figure 1 antibiotics-14-00850-f001:**
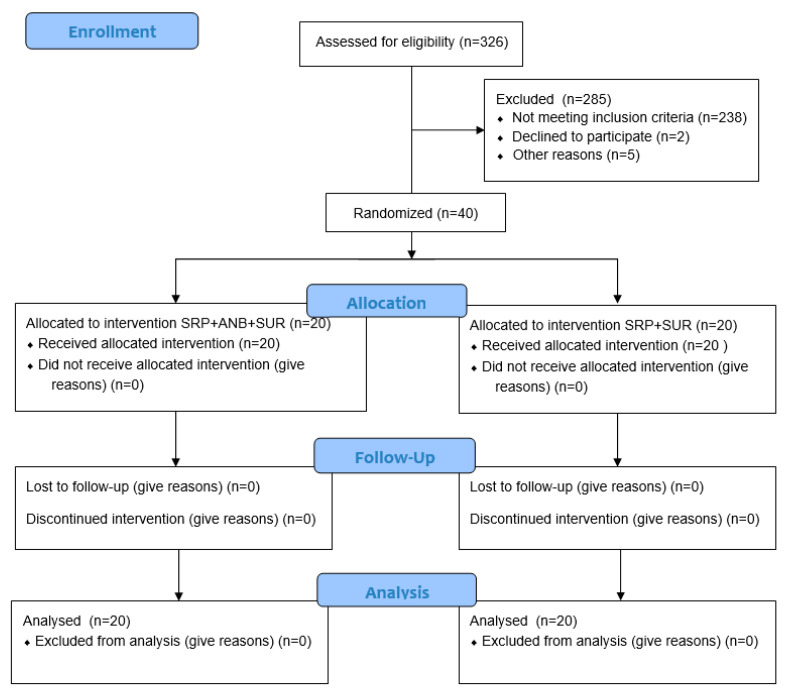
Study flowchart.

**Figure 2 antibiotics-14-00850-f002:**
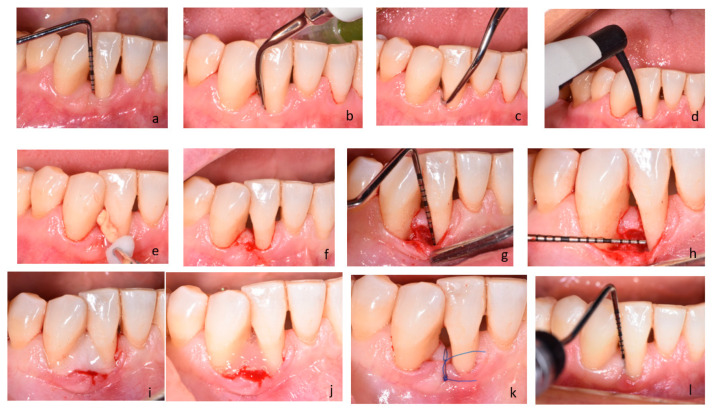
(**a**–**l**) Subject treated with SRP+ANB+MIST/M-MIST (test group): (**a**) probing pocket depth on distal surface of tooth 42; (**b**) SRP procedure two weeks before the surgery with ultrasound device; (**c**) SRP procedure two weeks before the surgery with hand instruments; (**d**) intra-pocket application of the antibiotic; (**e**) removal of excess material; (**f**) a surgical site, MPPT; (**g**) intrabony defect after flap exposure and debridement, vertical measurement; (**h**) intrabony defect, horizontal measurement; (**i**) EDTA application; (**j**) EMD application; (**k**) flap reposition and closure with non-absorbable sutures-frontal view; and (**l**) probing pocket depth on the surgical site 6 months post-op.

**Figure 3 antibiotics-14-00850-f003:**
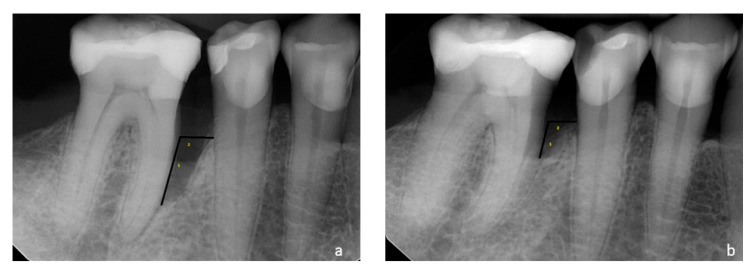
(**a**,**b**) Radiological measurements: 1. RIDD and 2. RIDW. (**a**) Radiograph of bone defect on mesial surface of tooth 46 before treatment; (**b**) radiograph 6 months after surgery.

**Table 1 antibiotics-14-00850-t001:** Characteristics of study participants and teeth included to the surgical procedures (clinical parameters in mm).

	Control Group	Test Group	Significance
No.	20	20	-
Gender	17F/3M	11F/9M	*p* * = 0.03
Mean age	44.7 (5.77)	49.45 (9.31)	NS (*p* ** = 0.06)
Incisors/canines/ premolars/molars	7/3/7/3	12/0/4/4	-
Surgical access	MPPT n = 13 SPPF n = 7	MPPT n = 18 SPPF n = 2	NS (*p* ** = 0.06)
Numbers of walls (1-wall/2-walls/3-walls)	1/15/4	0/16/4	-
Mean intrabony depth (IDD)	4.6 mm (±1.23)	4.87 mm (±1.66)	NS (*p* ** = 0.81)
Mean intrabony width (IDW)	2.7 mm (±0.65)	2.67 mm (±0.61)	NS (*p* ** = 0.98)

*p* *—Pearson’s Chi test; *p* **— Mann–Whitney’s U test; MPPT—modified papilla preservation technique, SPPF—simplified papilla preservation flap.

**Table 2 antibiotics-14-00850-t002:** (**A**,**B**) Clinical parameters in test and control groups before treatment and 6 months post op.

**(A)**				
	**2 Weeks Pre op.**	**Baseline**	**6 Months Post op.**	***p* ***
FMPS				
Test group	15.19% (±5.9)	12.35% (±5.19)	23.19% (±11.28)	0.001
Control group	15.6% (±5.48)	13.39% (±5.93)	17.03% (±8.56)	0.031
*p* **	NS (*p* ** = 0.86)	NS (*p* ** = 0.35)	NS (*p* ** = 0.054)	-
FMBOP				
Test group	13.66% (±5.28%)	12.14% (±5.32)	21.02% (±9.61)	NS
Control group	16.69% (±4.36%)	16.95% (±4.3)	19.78% (±12.42)	NS
*p* **	NS (*p* ** = 0.061)	0.003	NS (*p* ** = 0.473)	-
PI				
Test group	23.75% (±20.63)	15% (±14.95)	27.5% (±22.79)	0.022
Control group	17.5% (±16.42)	15% (±18.84)	16.25% (±23.33)	NS
*p* **	NS (*p* ** = 0.343)	NS (*p* ** = 0.821)	NS (*p* ** = 0.108)	-
BOP				
Test group	40% (±18.84)	30% (±17.39)	27.5% (±21.3)	NS
Control group	45% (±22.36)	33.75% (±18.62)	32.5% (±28.21)	0.047
*p* **	NS (*p* ** = 0.506)	NS (*p* ** = 0.446)	NS (*p* ** = 0.792)	-
(**B**)				
	**2 Weeks Pre op.**	**Baseline**	**6 Months Post op.**	***p* * **
PD				
Test group	6.8 mm (±0.95)	6.7 mm (±0.86)	3.8 mm (±0.69)	<0.001
Control group	6.95 mm (±0.88)	6.85% (±0.67)	3.95 mm (±1.09)	<0.001
*p* **	NS (*p* ** = 0.526)	NS (*p* ** = 0.414)	NS (*p* ** = 0.745)	
CAL				
Test group	8.2 mm (±1.98)	8.1 mm (±1.91)	5.85 mm (±1.98)	<0.001
Control group	7.5 mm (±1.14)	7.4 mm (±0.99)	4.55 mm (±1.79)	<0.001
*p* **	NS (*p* ** = 0.361)	NS (*p* ** = 0.335)	NS (*p* ** = 0.05)	
GR				
Test group	1.4 mm (±1.56)	1.4 mm (±1.56)	2.05 mm (±1.9)	<0.001
Control group	0.55 mm (±0.82)	0.55 mm (±0.82)	0.75 mm (±0.85)	NS
*p* **	NS (*p* ** = 0.078)	NS (*p* ** = 0.078)	0.021	
KT				
Test group	4.5 mm (±2.16)	4.6 mm (±2.13)	4.3 mm (±2.05)	NS
Control group	4.65 mm (±1.87)	4.65 mm (±1.87)	4.75 mm (±2.12)	NS
*p* **	NS (*p* ** = 0.857)	NS (*p* ** = 0.989)	NS (*p* ** = 0.563)	

*p* *— Friedman’s ANOVA test (changes in time); *p* **— Mann–Whitney U test (between groups); NS—non significant; FMPS—full-mouth plaque score; FMBOP—full-mouth bleeding on probing; PI—plaque index; BOP—bleeding on probing; PD—probing depth; CAL—clinical attachment level; GR—gingival recession; KT—keratinized tissue.

**Table 3 antibiotics-14-00850-t003:** Real-time polymerase chain reaction (PCR) test results in test and control groups before treatment and 3 months post op.

		2 Weeks Pre Op.	Baseline	*p* ***	3 Months Post Op.	*p* *
*Treponema denticola*	Test group	3.99 × 10^5^ (±9.36 × 10^5^)	1.17 × 10^4^ (±6.41 × 10^6^)	<0.001	9.93 × 10^5^ (±2.55 × 10^6^)	<0.001
	Control group	3.99 × 10^5^ (±9.36 × 10^5^)	2.28 × 10^5^ (±8.51 × 10^5^)	NS	7.69 × 10^5^ (±1.94 × 10^6^)	NS
*p* **		NS	NS	-	NS	-
*Fusobacterium nucleatum*	Test group	3.27 × 10^5^ (±8.83 × 10^5^)	2.68 × 10^4^ (±6.38 × 10^4^)	0.002	2.49 × 10^6^ (±1.07 × 10^7^)	0.001
	Control group	1.65 × 10^5^ (±3.77 × 10^5^)	7.38 × 10^4^ (±1.01 × 10^5^)	NS	8.7 × 10^4^ (±2.23 × 10^5^)	NS
*p* **		NS	0.019	-	NS	-
*Tannerella forsythia*	Test group	5.9 × 10^5^ (±1.49 × 10^6^)	1.17 × 10^4^ (±4.01 × 10^4^)	0.001	1.38 × 10^5^ (±3.59 × 10^5^)	0.021
	Control group	4.48 × 10^5^ (±8.18 × 10^5^)	2.33 × 10^5^ (±8.51 × 10^5^)	0.032	1.4 × 105 (±4.36 × 10^5^)	0.018
*p* **		NS	0.026	-	NS	-
*Porphyromonas gingivalis*	Test group	1.99 × 10^6^ (±4.95 × 10^6^)	1.12 × 10^5^ (±3.8 × 10^5^)	<0.001	9.37 × 10^5^ (±2.78 × 10^6^)	0.001
	Control group	1.31 × 10^6^ (±4.26 × 10^6^)	6.83 × 10^5^ (±2.43 × 10^6^)	0.008	1.68 × 10^5^ (±7.35 × 10^5^)	NS
*p* **		NS	NS	-	NS	-
*Prevotella intermedia*	Test group	7.15 × 10^3^ (±2.73 × 10^4^)	1.98 × 10^2^ (±5.65 × 10^2^)	0.009	9.86 × 10^4^ (±4.13 × 10^5^)	0.019
	Control group	2.11 × 10^5^ (±7.65 × 10^5^)	3.34 × 10^5^ (±1.26 × 10^6^)	NS	5.98 × 10^5^ (±2.57 × 10^6^)	NS
*p* **		NS	0.03	-	NS	-
*Aggregatibacter actinomycetemcomitans*	Test group	1.12 × 10^7^ (±3.47 × 10^7^)	2.92 × 10^6^ (±1.3 × 10^7^)	NS	3.96 × 10^6^ (±1.77 × 10^7^)	NS
	Control group	2.69 × 10^5^ (±8.59 × 10^5^)	9.41 × 10^5^ (±4.13 × 10^6^)	NS	1.38 × 10^5^ (±3.75 × 10^5^)	NS
*p* **		NS	NS	-	0.002	-

*p* *— Friedman’s ANOVA test (changes in time); *p* **— Mann–Whitney U test (between groups); *p* ***—Wilcoxon pair test (2 weeks pre op–6 months post op); NS—non significant.

**Table 4 antibiotics-14-00850-t004:** Radiological intrabony defect depth (R IDD) and width (R IDW) in test and control groups at baseline and 6 months post op.

	2 Weeks Pre Op.	6 Months Post Op.	*p* *** (2 Weeks Pre Op–6 Months Post Op)
R IDD (mm)			
Test group	3.53 (±0.92)	1.65 (±0.78)	<0.00
Control group	3.43 (±0.57)	1.57 (±1.05)	<0.00
*p* **	NS (*p* ** = 0.807)	NS (*p* ** = 0.87)	-
R IDW (mm)			
Test group	2.14 (±0.68)	1.77 (±0.84)	<0.00
Control group	1.96 (±0.76)	1.25 (±1.06)	<0.00
*p* **	NS (*p* ** = 0.506)	NS (*p* ** = 0.082)	-

*p* **— Mann–Whitney U test (between groups); *p* ***—Wilcoxon pair test (2 weeks pre op–6 months post op); NS—non significant; R IDD—radiographic intrabony defect depth; R IDW—radiographic intrabony defect width.

## Data Availability

The datasets used and/or analyzed during the current study are available from the corresponding author on reasonable request.

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
