# Peer review of "The Influence of Local Antibiotic Therapy on the Microbiological, Clinical, and Radiological Outcomes Following Minimally Invasive Periodontal Surgery in the Treatment of Intrabony Defects—A Randomized Clinical Trial"

_antibiotics, 2025, doi:10.3390/antibiotics14090850_

Round 1
Reviewer 1 Report
Comments and Suggestions for Authors
Thank you for this fine work. There are some minor issues to be covered:
- page 4 section Results, first sentence seems to be cut off some other paragraph. Please, check and rephrase.
- section Results may benefit in presentation with added subheadings similar as used in Materials and methods section, eg. “Periodontal clinical parameters”, “Microbiology findings “ and “Radiological findings”
- section title Materials and methods may be considered to be changed to “Subjects and procedures”
- page 14, lines 469-471, sentence “ The exclusion criteria…” should include list of systemic diseases affecting periodontal healing stated in brackets after that part of sentence eg. “(diabetes, … “ etc
Author Response
Dear Reviewer,
we would like to thank you for your thorough evaluation of our manuscript. We hope that the corrections and improvements introduced will meet with your acceptance and favor.
Comment: page 4 section Results, first sentence seems to be cut off some other paragraph. Please, check and rephrase.
Respond: Thank you for this comment. We moved the sentence.
Comment: section Results may benefit in presentation with added subheadings similar as used in Materials and methods section, eg. “Periodontal clinical parameters”, “Microbiology findings “ and “Radiological findings”
Respond: Thank you for this comment. We added subheadings
Comment: section title Materials and methods may be considered to be changed to “Subjects and procedures”
Respond: Thank you for this comment. We prefer to leave the title Material and method as it is guidelined in Research Manuscript Sections
Comment: page 14, lines 469-471, sentence “ The exclusion criteria…” should include list of systemic diseases affecting periodontal healing stated in brackets after that part of sentence eg. “(diabetes, … “ etc
Respond: Thank you for this comment. Added.

Reviewer 2 Report
Comments and Suggestions for Authors
Overall Assessment
The study addresses an important clinical topic: the adjunctive role of local antibiotic therapy in minimally invasive periodontal regeneration procedures. While the study design is interesting, there are several areas requiring substantial improvement, including study rationale, methodological clarity, statistical analysis, and the depth of the discussion.
Abstract:
-
The abstract is too concise and lacks detail regarding the methodology, statistical power, and limitations.
-
The phrase "at to 6 months" (line 33) contains a grammatical error and should be corrected to "up to 6 months".
-
The conclusion section of the abstract is vague; it should emphasize the clinical significance of findings and practical recommendations.
Recommendation: Expand the abstract to include key numerical findings (e.g., mean changes in PD, CAL), p-values, and confidence intervals for clarity.
Introduction
-
The introduction provides adequate background on periodontitis and local drug delivery systems but could be streamlined to avoid excessive citations from lines 59–139.
-
The novelty of the study is not clearly stated. The authors should explicitly explain how this study adds new evidence compared to previous research (e.g., studies using MIST with local antibiotics).
-
There are grammatical errors and awkward phrases, e.g., “focus was on assessing the effectiveness of systemic antibiotic therapy as an additional element in non-surgical treatment” (line 82). Consider revising for clarity.
Recommendation: Add a clear research gap and hypothesis at the end of the introduction.
Methods
-
Sample size calculation (lines 442–448): The calculation is presented but lacks details on expected effect size and variance references.
-
Randomization: The method of randomization is vaguely described. Was allocation concealment or block randomization performed?
-
Blinding: The study states that neither patient nor surgeon was blinded (line 552). This introduces bias. Were assessors blinded to treatment groups?
-
Inclusion/exclusion criteria: These are comprehensive but could be better structured as a table for clarity.
-
Microbiological analysis: PCR analysis is mentioned, but the DNA extraction protocol and quality control measures are not sufficiently described.
Results
-
Results are well-organized with tables, but Table 1 and Table 2 are overly dense and could be split into sub-tables (e.g., demographics vs. clinical parameters).
-
The narrative repeats table data verbatim instead of highlighting key trends or statistical differences.
-
Microbiological results (lines 181–205) require clearer reporting of absolute vs. relative changes in bacterial counts.
-
There is inconsistent use of decimal separators (e.g., “4,6mm” vs. “4.6 mm”).
Recommendation: Use consistent SI formatting and focus on clinically relevant outcomes rather than excessive raw data.
Discussion
-
The discussion is lengthy but lacks depth in critical analysis. While many studies are cited, the clinical relevance and limitations of the current findings are not adequately synthesized.
-
The authors mention EMD’s antibacterial effects (lines 381–395) but do not critically evaluate how this might confound the study’s primary objective of assessing local antibiotic adjunctive effects.
-
The limitations section (lines 420–439) is strong but could expand on potential bias introduced by the lack of blinding and small sample size.
Recommendation: Condense repetitive literature review, provide a focused interpretation of results, and expand on clinical implications (e.g., cost-effectiveness of local antibiotics).
Language and Style
-
The manuscript has multiple grammatical and typographical errors (e.g., “our study” instead of “our study” at several locations, lines 307, 333, 346
-
English editing is required for readability.
Figures and Tables
-
Figures (flowchart, clinical cases) are helpful but need higher resolution and standardized labeling.
-
Tables should include footnotes for all abbreviations (e.g., FMPS, FMBOP, CAL, PD).
References
should be up to date, and Journal names should be standardized according to Antibiotics journal style.
Author Response
Dear Reviewer,
we would like to thank you for your thorough evaluation of our manuscript. We hope that the corrections and improvements introduced will meet with your acceptance and favor.
Abstract:
Comment: The abstract is too concise and lacks detail regarding the methodology, statistical power, and limitations. The conclusion section of the abstract is vague; it should emphasize the clinical significance of findings and practical recommendations.
Recommendation: Expand the abstract to include key numerical findings (e.g., mean changes in PD, CAL), p-values, and confidence intervals for clarity.
Respond: Thank you for this comment. We would like to expand on the abstract, in order to add the mentioned elements, but we must adhere to the word limits in this section (around 250 words). Elements mentioned in comment are included in the manuscript.
Comment: The phrase "at to 6 months" (line 33) contains a grammatical error and should be corrected to "up to 6 months".
Respond: Thank you for this comment. We corrected the error.
Introduction
Comment: The introduction provides adequate background on periodontitis and local drug delivery systems but could be streamlined to avoid excessive citations from lines 59–139.
Respond: Thank you for this comment. We have rebuilt the text.
Comment: The novelty of the study is not clearly stated. The authors should explicitly explain how this study adds new evidence compared to previous research (e.g., studies using MIST with local antibiotics).
Respond: Thank you for this comment. We have put it in a more precise sentence
Comment: There are grammatical errors and awkward phrases, e.g., “focus was on assessing the effectiveness of systemic antibiotic therapy as an additional element in non-surgical treatment” (line 82). Consider revising for clarity.
Respond: Thank you for this comment. We have rebuilt the text
Comment: Recommendation: Add a clear research gap and hypothesis at the end of the introduction.
Respond: Thank you for this comment. Added
Methods
Comment: Sample size calculation (lines 442–448): The calculation is presented but lacks details on expected effect size and variance references.
Respond: Thank you for this comment. In the sample size paragraph we included the above-mentioned elements: clinically significant difference of 1.0 mm; type I error was set at 0.05 level and power at 0.80; standard deviation (SD) was 1.0 mm; the SD value is mentioned, and SD is the square root of the variance.
Comment: Randomization: The method of randomization is vaguely described. Was allocation concealment or block randomization performed?
Respond: Allocation of patients to test (SRP+ANB+SUR) and control (SRP+SUR) groups was performed according to a randomization table prepared by a statistician. When a new patient diagnosed with periodontitis stage III was included to the study, he was randomly assigned to which group he would be placed by an operator A.S.
Comment: Blinding: The study states that neither patient nor surgeon was blinded (line 552). This introduces bias. Were assessors blinded to treatment groups?
Respond: Thank you for this comment. There was one assessor (A.B.) and she was blinded.
Comment: Inclusion/exclusion criteria: These are comprehensive but could be better structured as a table for clarity.
Respond: Thank you for this comment. We would prefer to leave this in the main text, as we have divided remaining tables.
Comment: Microbiological analysis: PCR analysis is mentioned, but the DNA extraction protocol and quality control measures are not sufficiently described.
Respond: Thank you for this comment. We included required information.
Results
Comment: Results are well-organized with tables, but Table 1 and Table 2 are overly dense and could be split into sub-tables (e.g., demographics vs. clinical parameters).
Respond: Thank you for this comment. We have rebuilt tables.
Comment: The narrative repeats table data verbatim instead of highlighting key trends or statistical differences.
Respond: Thank you for this comment. We have rebuilt the text.
Comment: Microbiological results (lines 181–205) require clearer reporting of absolute vs. relative changes in bacterial counts.
Respond: Thank you for this comment. We did the corrections.
Comment: There is inconsistent use of decimal separators (e.g., “4,6mm” vs. “4.6 mm”).
Respond: Thank you for this comment. Corrected
Comment: Recommendation: Use consistent SI formatting and focus on clinically relevant outcomes rather than excessive raw data.
Respond: Thank you for this comment. Corrected
Discussion
Comment: The authors mention EMD’s antibacterial effects (lines 381–395) but do not critically evaluate how this might confound the study’s primary objective of assessing local antibiotic adjunctive effects.
Respond: Thank you for this comment. We have included this in limitations
Comment: The limitations section (lines 420–439) is strong but could expand on potential bias introduced by the lack of blinding and small sample size.
Respond: Thank you for this comment. We have added this in limitations
Comment: Recommendation: Condense repetitive literature review, provide a focused interpretation of results, and expand on clinical implications (e.g., cost-effectiveness of local antibiotics).
Respond: Thank you for this comment. We have corrected and expanded the discussion.
Language and Style
Comment: The manuscript has multiple grammatical and typographical errors (e.g., “our study” instead of “our study” at several locations, lines 307, 333, 346
English editing is required for readability.
Respond: Thank you for this comment. The text was proofread by a professional translator.
Figures and Tables
Comment: Figures (flowchart, clinical cases) are helpful but need higher resolution and standardized labeling.
Respond: Thank you for this comment. We have sent original files.
Comment: Tables should include footnotes for all abbreviations (e.g., FMPS, FMBOP, CAL, PD).
Respond: Thank you for this comment. Corrected
References
Comment: Should be up to date, and Journal names should be standardized according to Antibiotics journal style.
Respond: Thank you for this comment. Corrected
